# Textile Materials Modified with Stimuli-Responsive Drug Carrier for Skin Topical and Transdermal Delivery

**DOI:** 10.3390/ma14040930

**Published:** 2021-02-16

**Authors:** Daniela Atanasova, Desislava Staneva, Ivo Grabchev

**Affiliations:** 1Department of Textile and Leathers, University of Chemical Technology and Metallurgy, 1756 Sofia, Bulgaria; d.atanasova1@abv.bg; 2Faculty of Medicine, Sofia University “St. Kliment Ohridski”, 1407 Sofia, Bulgaria; i.grabchev@chem.uni-sofia.bg

**Keywords:** transdermal therapy, smart textile, drug delivery, stimuli-responsive polymer

## Abstract

Textile materials, as a suitable matrix for different active substances facilitating their gradual release, can have an important role in skin topical or transdermal therapy. Characterized by compositional and structural variety, those materials readily meet the requirements for applications in specific therapies. Aromatherapy, antimicrobial substances and painkillers, hormone therapy, psoriasis treatment, atopic dermatitis, melanoma, etc., are some of the areas where textiles can be used as carriers. There are versatile optional methods for loading the biologically active substances onto textile materials. The oldest ones are by exhaustion, spraying, and a pad-dry-cure method. Another widespread method is the microencapsulation. The modification of textile materials with stimuli-responsive polymers is a perspective route to obtaining new textiles of improved multifunctional properties and intelligent response. In recent years, research has focused on new structures such as dendrimers, polymer micelles, liposomes, polymer nanoparticles, and hydrogels. Numerous functional groups and the ability to encapsulate different substances define dendrimer molecules as promising carriers for drug delivery. Hydrogels are also high molecular hydrophilic structures that can be used to modify textile material. They absorb a large amount of water or biological fluids and can support the delivery of medicines. These characteristics correspond to one of the current trends in the development of materials used in transdermal therapy, namely production of intelligent materials, i.e., such that allow controlled concentration and time delivery of the active substance and simultaneous visualization of the process, which can only be achieved with appropriate and purposeful modification of the textile material.

## 1. Introduction

The economic prosperity of a certain country is determined by the efforts this country puts into improving the quality of life of its population that largely depends on developing and implementing new technologies in *daily routine*. The progress of medicine and healthcare has led to prolonged average life expectancy of modern men. That, however, is constantly increasing peoples’ requirements for information and comfort. Therefore, nowadays, development of nanotechnology, electronic devices, wireless communication, and information technologies is increasingly entering areas that until recently were considered traditional, difficult to change, and rarely associated with new discoveries. These include some of the oldest man-made materials, such as textiles. Today, the word design is more relevant to them than ever before, because smart textiles have a great future utilization in healthcare [1,2,3], medicine [4,5,6,7,8], transport [9,10], sports and leisure [11,12], safety and personal protective equipment [13], construction [14], interior design [15,16], agriculture [17], sensors and biosensors [18,19,20], etc. Materials creation is associated with combining their known properties with new functionality to ensure active interaction with the environment, i.e., ability to react and adapt to changes [21].

Textile materials used for medical purposes are structures and products that are used in first aid, in clinical and hygienic practice. This usage in various forms dates back to ancient times. For example, natural materials such as silk, cotton, linen, etc., have been applied as dressings for wounds and sutures. In the 20th century, new in vivo application of biomedical textiles was established in the cardiovascular, digestive, excretory, etc., systems following the introduction of artificial and synthetic fibers. Today, the new challenges are focused in the field of regenerative and tissue engineering, in systems for delivery of biologically active substances (BAS), in the creation of intelligent textile materials for monitoring and treatment of various health conditions [22,23].

Because of their compositional and structural diversity, textiles are able to meet successfully all the requirements for specific medical treatment. The traditional characteristic properties of textile materials are flexibility; lightness; porosity; air permeability; hydrophilicity or hydrophobicity, depending on their composition. The surface and functional groups they have are suitable for modification with different systems for delivery of BAS. Thus, one can achieve controlled drug discharge depending on a number of factors related to the change in health status.

Currently, the most commonly used methods for BAS delivery in the human body are: peroral; intravenously; by inhalation; etc. In recent years, advances in drug delivery systems have given new impetus to another long-established mode of prevention and treatment, namely transdermal therapy. It facilitates effects upon a number of nutritional deficiencies that leads to problems in the immune, hormonal, and nervous systems, to protect cells from oxidative destruction, to influence the formation of tumors and therapy in patients with diabetes. Textiles modified with a stimuli-responsive drug carrier have great potential to answer to the specific patient case and can be used as both topical and systemic (transdermal) drug delivery systems [24]. Aromatherapy [25], antimicrobials [26] and painkillers [27], hormone therapy [28], treatment of acne [29] psoriasis [30], atopic dermatitis [31,32], melanoma [33], etc., are just some of the areas where textiles are involved as an important functional element of systems for transdermal therapy, wound dressings, dermatology, etc. The utilization of biomedical textiles in dermatology and transdermal therapy is characterized by easy application, greater comfort and less pain for the patients, their shorter hospitalization, etc. [34,35].

Several excellent review articles describing applications of textile materials and different pharmaceutical nanocariers in topical and transdermal drug delivery have been published [36,37,38]. The increased number of studies in this field is as a result of the current interest in the development of the platform for continuous monitoring and assisting the health status of individuals via telemedicine [39] and personalized medicine [40]. This modern concept for continuous personalized patient care influences the advance of wearable textiles that allow administering a predetermined drug dose at certain intervals. As the processes taking place in wound healing or transdermal therapy are complex, the design of delivery systems for BAS is dependent on a number of multi-stage, interconnected factors. One of the crucial factors to be considered is the ability of known nanocarriers to respond to external stimuli so that the delivery to the target and release control be more precise.

One of the main drawbacks of the controllable drug delivery is the initial burst release of drugs. The simple dip-coating of textile materials forms weak physical bonds between the functional groups of fibers and drugs. Their easy breaking is the reason for the significant release of drugs with a lack of control. The presence of suitable functional groups in the textile fabric can be used for chemical bonding with the drug carrier. In this way, textiles can be actively involved in the design of composite materials for drug delivery. The role of textile can be transformed from a passive drug matrix into an active participant, influencing the mechanism of drug release. Lis et al. have found that, if the complexes of essential oil with β-cyclodextrin are applied and crosslinking onto the cotton or polyester fabric, the bioactive substance release will be prolonged for hours. The observed difference between cotton and polyester fabrics shows that, the textile matrix affects directly the release mechanism [41].

The aim of this review is to present the new trends in the development of systems for skin topical and transdermal delivery of BAS wherein the textile materials are modified with different stimuli-responsive polymeric carriers. The survey also covers certain successful examples for them in literature, the used preparation and characterization methods, the elaboration and application problems, as well as prospects for their overcoming, some guidelines for the future design associated with multifunctionality and intelligent behavior.

## 2. Comparative Analysis of Peroral and Transdermal Medication

The delivery of BAS is an important aspect of medical practice and in realizing the healing process. Depending on the disease and on the condition of medicine, there are different forms of application (tablets, powders, drops, capsules, injections, etc.). In many cases and situations, however, transdermal administration (through the skin) may be much more appropriate. When taken orally, the drugs are absorbed in the stomach or intestinal tract. There, they participate in various metabolic processes in the course of which BAS lose their effective properties before being able to reach the medical goals. Therefore, much higher doses are needed to achieve the desired effect, which can lead to toxic metabolites in the liver [34,35]. The delivery of drugs through the skin bypasses the passage through the liver, which may reduce the doses administered. In addition, there are situations in which oral administration is difficult or even inapplicable-in cases of young children, people with swallowing problems, or of people with dementia. Then, transdermal therapy is a very easy and convenient treatment approach, since the skin provides a large area for drug delivery and is easily accessible; discomfort and physical pain are avoided; the patient enjoys comfort; his/her hospital stay and treatment costs are lower, etc. The development of various transdermal delivery systems dates a long time ago. Ointments and creams are the earliest skin topical and transdermal formulations. The disadvantages of these dosage forms are the requirement for repeated application by the patient, the variability of the dose, and possible dangerous administration and uncontrollability, e.g., the application of nitroglycerine in an ointment form has advantages over the parenteral or sublingual administration as sustained release and concentration. However, its drawback is the difficult controllability of the precise quantity of the cream applied, the layer thickness, and the treated skin area. The frequent repetition of the application over time also leads to inconvenience. Another disadvantage is a possible skin-to-skin transfer from patients to other people being in close contact with them. These shortcomings are among the reasons that led to the creation and development of patches as another way for transdermal drug delivery [42].

The market offers various transdermal patches as commercial products. They are multilayer systems in which the drug is released gradually, passing through a permeable membrane [43]. The early design of such patches included a reservoir containing liquid or gel drug, a backing layer, a membrane that controls the rate of drug release, and an adhesive layer. They are known as reservoir/membrane patches. Matrix patches are the next-generation. Their design includes a solution or suspension of active ingredient within a polymer or textile pad that is held in direct contact with the skin. Later, patches combined the drug and adhesive in a single layer. Those are lighter, thinner, more flexible patches and more comfortable to wear. To improve the properties of the patches, textile materials were also included in their design. A nonwoven polyester fabric has been used as a support for the other layers in the proposed patches [44]. The active patches are a new trend in transdermal delivery. Their combination with microneedles leads to drug delivery enhancement [42,43]. Problems with the application of transdermal patches are associated with their structure and ingredients (drug, excipients, and presence of metals (e.g., aluminium) in the backing layer). Some drawbacks can be skin irritation, allergic reactions, the possibility of dangerous drug leakage, overdose in improper use. The further patches development and, in general, transdermal drug delivery systems must overcome these problems using appropriate technologies and new forms of delivery.

## 3. Skin Structure. Mechanism of the Transdermal System for BAS Delivery. Factors Determining Its Effectiveness

Skin is the largest organ in the human body (about 2 m^2^) and performs various important functions, one of which is to protect the body from external factors (e.g., pathogens). It consists of three main layers: epidermis, dermis, and hypodermis. The epidermis, in particular, the *stratum corneum*, acts as a major barrier to the absorption of BAS. It is a lipophilic membrane of which thickness varies from 0.06 mm in the eyelids to 0.8 mm in the skin of the heels [45]. The dermis is the second deepest skin layer and is composed mainly of connective tissue. It is rich in blood vessels, through which the oxygen and nutrients necessary for its vital activity reach the skin. The subcutaneous tissue is under the dermis and is composed mainly of adipose tissue, with varying severity in different parts of the body. This layer helps regulate the temperature. Figure 1 shows the structure of skin and the routes BAS pass through it. The applied BAS must pass through lipophilic and hydrophilic membranes on the way to the dermis. The first barrier is the outermost layer of the epidermis also called the *stratum corneum*. It is 10–20 μm thick [46]. It consists of flat cells made up of keratin, which have turned into overlapping scales, arranged in many deposits (on the feet more than 100) embedded in a lipid matrix consisting mainly of ceramides, cholesterol, and free fatty acids. The *stratum corneum* is water-resistant, while acidic solutions and basic solutions make it swell and become permeable [47,48].

The skin condition difference, the integrity of its layers, the type of disease, and the necessary treatment are some of the factors that determine the technology for creating pharmaceutical materials. In transdermal therapy, BAS must pass through all layers of the skin to reach the systemic circulation. In the topical skin treatment, BAS aim at treating the skin itself and its layers on a particular affected body area (epidermis and dermis) [49]. In this case, the normal skin structure and function may be preserved or disrupted to a greater or lesser extent. The skin injury is known as a wound. According to the number of affected skin layers, wounds are: a superficial wound (only epidermal layer is affected); a partial thickness wound with injures, involving both the epidermis and the deeper dermal layers, the blood vessels, sweat glands, and hair follicles, inclusive; full-thickness wounds, the underlying subcutaneous fat or deeper tissues are damaged in addition to the epidermis and dermal layer [50].

The three main routes that BAS can penetrate through the *stratum corneum* are intercellularly, intracellularly, and via the follicles. The intercellular pathway predominates over the intracellular [49]. Hydrophobic drugs penetrate skin intercellular. They dissolve and diffuse through the non-aqueous lipid matrix of the *stratum corneum* [51]. Hydrophilic molecules pass through the intracellular pathway penetrating through the corneocyte cells. However, the molecules need to pass through the intercellular lipids to reach the next corneocyte. Therefore, a part of their pathway is partially intercellular and may be decisive for the diffusion rate.

For optimum absorption, BAS should have a balanced aqueous and lipid solubility in order to permeate to transverse stratum corneum and underlying aqueous layer. Drugs with high partition coefficient P (octanol/water) are not ready to leave the lipid portion of skin and drugs with low P will not permeate. The logP between 1 and 3 is required as the most molecules pass the *stratum corneum* by both routes [52]. Low molecular weight (MW < 500 Da) and high pharmacological potency [53] are other drug molecules characteristics important for a transdermal therapy.

The third possible route for hydrophilic compounds to reach the dermis is through defects in the skin structure—for instance, through hair follicles, sweat glands, etc. [54]. The transfer through the follicle is considered insignificant, as according to data, the part of the skin covered with hair is quite small—only 0.1%.

Individuals absorb BAS to a different extent because many other factors affect the transdermal drug delivery, namely: skin condition (hydration, subcutaneous diseases or injuries); thickness of the *stratum corneum*; skin metabolism; melanin content; blood supply; age; ethnic differences; body temperature; contact time, etc.

## 4. Biofunctional Textile Materials and Their Preparative Methods

The development and use of biofunctional textiles is a rapidly developing interdisciplinary field of science and application, due to the diverse possibilities of use and growing demand. The contact of the material with human skin is essential, so the new properties obtained must be effective, but also provide comfort to the user. These textiles include materials with antimicrobial properties, those that supply BAS or absorbents for substances released during sweating and leading to an unpleasant odor, etc.

The manufacturing process of biofunctional textile materials consists of fibers production using different natural or synthetic polymers and yarn processing by various engineering methods. The next step is surface modification via different chemical, physical, and biological treatment [55]. Biofunctional fabrics combine conventional textiles with modern advances in the development of drug delivery systems, as well as the application of new approaches to the transdermal application of bioactive molecules [56,57,58,59]. Direct immobilization of the active compound on the fabric surface can lead to some disadvantages as a change in its biological activity, controllability, and problems with skin interaction (Figure 2A). Therefore, the tendency is to encapsulate BAS into a suitable carrier that meets certain requirements for its controlled release and action (Figure 2B) [57,59]. Other methods involve surface modification of textile materials and introduction of new functional groups, as well as implementation of polymeric binders for immobilization or encapsulation of various organic and inorganic micro- and nanoparticles (Figure 2C). In all three cases, the drug carrier may be physically deposited on the surface of the textile material or covalently bonded to it, or to the polymeric material, as in the case of Figure 2C [59].

Biofunctional textile materials can be produced by traditional techniques: exhaustion from solution; irrigation; spraying; pad-dry-cure; using sol-gel or layer-by-layer technique [55,60]. They can be classified according to their chemical composition, structure, and the way drug carriers attach. The materials used in medical practice can be produced from only one fiber type or from different types of fibers. The choice depends on the desired specific characteristics of the obtained material, which combine the medical purposes with appropriate functional properties, providing comfort for the patient [61]. For example, viscose/polyester dressings integrate the properties of viscose as facile modification with those of drug carriers: high absorbency, breathability, comfort, and softness with the resistance of wrinkle, tear, and microbial attacks of strong polyester fibers [62]. The individual fibers or threads can be loaded or modified with drug carriers. After that, they are assembled into a flexible wound dressing or other biofunctional material through textile engineering processes. This approach was used to prepare cotton threads coated with a layer of electrically conductive ink and covered with a hydrogel layer of alginate/poly(ethylene glycol) diacrylate carrying thermoresponsive particles. The fibers were then assembled into fabrics for delivering antibiotics and vascular endothelial growth factor [63].

The structure of fabrics can be woven, knitted, and nonwoven. The woven fabric can be modified with bioactive complex that are physically absorbed or adsorbed, coated, encapsulated or covalently conjugated on the fabric [41]. Nonwoven medical textiles have highly porous structure and adequate compression behavior that enhance the healing rate by controlled bioactive agent delivery, transport of nutrients, cellular migration, and metabolic wastes to allow the regeneration and the formation of new tissue [64]. The diameter of the fibers and the structure of the knitted fabric have been shown to play an important control role in the drug-releasing properties of the material [65]. The interaction of the transdermal delivery system with all conventional textile materials leads to the production of wearable devices for the delivery of drugs or biofunctional clothing.

The stimuli-responsive polymers can be attached to the textile surface by physical or chemical bonds. The physical adsorption can be used when the aim is to release the whole bioactive complex from stimuli-responsive polymer and conjugated BAS. In this case, the stimuli trigger the disruption of low energy physical bonds. When the corresponding stimulus-responsive polymer is bonded covalently to the functional groups of the fibers, the role of the applied stimuli is to change the structure of the polymer and to stimulate the release of BAS alone.

New treatment methods—low temperature plasma, corona, light curing and photopolymerization—have been applied successfully to improve the interaction of stimuli-responsive polymers with fiber surface. These methods produce effects in a depth of a few nanometers and can be used for changing the surface specifics of textile fibers without any change in bulk properties [66].

## 5. Requirements to Be Met by BAS Delivering Systems and Biofunctional Textiles

The main requirements for BAS delivery systems are biocompatibility and controllability. Important elements of biocompatibility are toxicity, carcinogenicity, mutagenicity, teratogenicity. In the case of biodegradable materials, not only the material itself must be safe, but also its degradation products. Therefore, the use of biofunctional tissues should not impact negatively the skin condition and produce side effects.

Another important aspect of BAS delivery systems is the optimal content and sustained release of therapeutics, cosmetics, or imaging agents. Any overdose or too low concentration may cause various side effects. In the first case, skin irritations, erythema, dryness, and peeling, as well as allergic reactions associated with rashes, etc., may occur. In the second case, side effect may be related to problems in the production of composite materials, associated with low efficiency of BAS loading or with its preliminary release in the process of preparation of the material itself [67]. The drug release depends on the degree of fabric loading and it is triggered by a series of physiological factors, like enzymes, hydro-electrolytic secretion of the sweat glands with a slight acid pH (coetaneous pH = 5.5) that can favor the drug release through diffusion, body temperature, and skin friction with the fabric. The external factors as light, magnetic field, ultrasound, and electrical stimuli can be used to influence sustainable BAS release [68,69].

On the other hand, biofunctional textiles, depending on the conditions of use, must remain stable for the delivery period and then be recycled or disposed of properly. Most drug delivery systems are relatively expensive, which makes the resulting biofunctional textiles less affordable.

## 6. Instrumental/Analytical Techniques Used for Characterization of Biofunctional Textiles

Biofunctional textiles are generally characterized as complex material for their morphology, and mechanical properties [41,70]. Their functional properties include drug loading efficiency and release, degradation if applied. Various analytical techniques are utilized to evaluate their development and application, in vitro and in vivo studies, inclusive. The instrumental/analytical techniques used for characterization of biofunctional textiles can be divided into two groups. The techniques from the first group aim at characterizing the efficiency of functionalization treatment and exploitation properties of textile material. Those from the second group are analyses of the pharmaceutical properties of a composite material and its possible influence on the human. The analytical methods used are: colorimetric and fluorescent analysis; Fourier-transform infrared and Raman spectroscopy for monitoring the modification of textile materials; scanning electron microscope (SEM) and transmission electron microscope (TEM) for morphology and configuration of polymeric materials and BAS; thermogravimetric analysis (TGA) and differential scanning calorimetry (DSC) analysis; X-ray photoelectron spectroscopy (XPS), X-ray diffraction analysis, and energy-dispersive X-ray spectroscopy for superficial characterization of the materials and for determining the interaction between the individual elements. Mechanical tests may be required to assess the stability of the textile according to the conditions of use.

Quantification of drug loading efficiency and drug-releasing is evaluated by suitable analytical methods. First, drug extracting is assessed using solvent or phosphate-buffered saline (PBS) at 37 °C. The next step is resorting one of following quantification methods (spectral methods, high-performance liquid chromatography (HPLC), gel permeation chromatography (GPC), and mass spectroscopy). The transdermal efficacy of biofunctional textiles for in vitro drug release is performed using Franz-type diffusion cells [70]. The cytotoxicity of textile materials used in topical and transdermal skin therapy is assessed by testing the viability of skin keratinocytes.

## 7. Stimuli-Responsive Drug Carrier for Delivering Bioactive Substances

Microencapsulation is a process by which the active substance is coated with a polymeric material called a shell. Microcapsules immobilized onto textile materials find various applications, including the supply of BAS. In recent years, however, active studies have been carried out on other structures, which have shown very good properties with regard to in vivo delivery of drugs that is promising for their in vitro application. The review by M.R. ten Breteler et al. [71] deals with implementation of textile materials modified with cyclodextrins, aza-crown ethers, fullerenes, ion exchange or hollow fibers, as well as with nanoparticles used as BAS carriers for treating acne, psoriasis, atopic dermatitis, melanoma, etc. [72]. Massella et al. expand the possible drug carriers for textile modification with nanospheres, micro- and nanocapsules, liposomes, inorganic particles, and micro-hydrogel [36]. In addition to the above structures, dendrimers are other interesting polymeric forms for biofunctional tissue production [73]. The present review describes some of vanguard polymeric formations as dendrimers, polymer micelles, liposomes, and various polymer nanocomposites, as well as crosslinked polymers in the form of hydrogels (Figure 3). What is common of the considered systems for BAS delivery is their intelligence (ability to react to various internal and external influences) which allows controlled BAS release. Thus, BAS use for application to the skin and in transdermal therapy becomes safer and more successful. The release mechanism of these systems includes desorption of BAS, diffusion through the carrier matrix, erosion, biodegradation, etc. Stimuli-responsive polymers are “intelligent” materials that can respond to a different small external stimulus with a change in their basic properties. They can be classified according to their nature as physical (temperature, ultrasound, mechanical stress, irradiation with light, magnetic and electrical fields), chemical (variation of pH and ions, dipole-dipole interaction, polarity of environment), and biological (BAS, biomolecules, proteins, enzymes) [74,75]. In shape, these materials can be linear, cross-linked, or branched.

### 7.1. Dendrimers

Dendrimers are highly branched, star-shaped macromolecules composed of three components: a central core, an inner dendritic structure (branches), and an outer surface with a large number of functional groups [76,77,78]. The physical characteristics, as well as their monodispersity, water-solubility, encapsulation ability, large number of functional peripheral groups, and low cytotoxicity, make those macromolecules suitable candidates as BAS carriers [79]. So far, dendrimers have been finding a variety of applications in textile engineering, including surface modification and finishing, drug delivery, improvement of dyeing, and permanent attachment to the fabric of fragrance, etc. [80]. Many times dendrimers have already been used successfully as pharmaceutical excipients [81]. They have a complex role including improved drug solubility and delivery, sensor properties. Their acting as penetration enhancers is very important for transdermal therapy [82]. There are three possible approaches to dendrimer loading [81]. According to the first, the active compound is covalently bound to the periphery of the dendrimer. The second involves interaction of the compound with the functional groups in the branches through ionic or coordination interactions, while in the case of the third approach, the dendrimer acts as a micelle and encapsulates the compound. This approach has been used to encapsulate various hydrophobic compounds and deliver them as anticancer therapy [83]. Studies have shown that PAMAM dendrimers can effectively facilitate the penetration of non-steroidal anti-inflammatory drugs (Ketoprofen and Diflunisal) through the skin [84]. Chauhan etc. demonstrated that the PAMAM dendrimer is effective as a transdermal drug-delivery system and increased the flux of indomethacin across the skin in vitro, as well as in vivo [85].

Different stimuli can direct the reaction of a dendrimer drug carrier. This “smart” behavior of dendrimers and their respond to more than two stimuli can lead to significant improvement in diagnosis, drug delivery, therapy, and effectiveness [86]. It has been found that, size, surface charge, and hydrophobicity affect the penetration pathway and the efficiency of dendrimer movement through the skin layers. Second generations of PAMAM dendrimers have better skin penetration than that of dendrimers of higher generations. Carboxyl and acetyl-terminated dendrimers go through the skin layers by intercellular pathway. In contrast, amine-terminated dendrimers internalized into individual cells both in the epidermal and dermal layers. This increased uptake leads to a higher accumulation of the dendrimer in the skin layers, which makes them potential candidates for localized treatment of skin diseases. The additional modification of the surface dendrimer groups leading to higher hydrophobicity and to partition coefficient P resulting in increased skin absorption and retention [87]. The wide range of stimuli, either endogenous (acid, enzyme, and redox potentials) or exogenous (light, ultrasound, and temperature change) allows great flexibility in the design of stimuli-responsive dendrimers. This design of intelligent dendrimers permits the delivery of BAS to be controlled spatially, temporally, and quantitatively according to the specific therapy [88].

### 7.2. Polymeric Micelles

Micelles range 5 to 100 nm in size and are composed of amphiphilic polymers or surfactants that agglomerate in an aqueous medium above a certain concentration, called ‘critical micelle formation concentration’ [89]. Van der Waals bonds arise between the hydrophobic groups and this energy causes the micelles formation. Polymer micelles have a wide range of applications, such as the supply of anticancer agents for tumor therapy, for treatment of neurodegenerative diseases, the supply of antifungal agents, in controlled delivery of genes, etc. [90]. In their review, Yu et al. have shown that, the polymeric micelles with intelligent response to different stimuli can be a promising approach for multi-drug co-delivery and control release [91]. Their loading onto textile materials may allow obtaining in the future materials for successful local or transdermal treatment.

### 7.3. Liposomes

Liposomes are used as biocompatible carriers of various BAS for pharmaceutical, cosmetic, and biochemical purposes. They have also been applied successfully in numerous textile processes. Having specific properties makes them valuable textile auxiliaries in the processes of preliminary preparation (washing, scouring, and bleaching), dyeing, and finishing of textile materials [92]. They consist of phospholipids, which are amphiphilic molecules that have a hydrophilic head and a non-polar hydrophobic tail. In an aqueous medium, phospholipids self-organize into bilayer membranes and form spherical hollow shapes with a liquid core [93]. This amphiphilic structure facilitates inclusion of hydrophilic molecules into the nucleus and hydrophobic molecules into the lipid membrane. Depending on the size and number of layers, liposomes are divided into: multilayer (of about 5–10 µm in size), large single-layer with a diameter of 100–200 nm, and small single-layer with a diameter of 20–50 nm. Depending on their encapsulating and sustaining release properties, they can be used in cosmetotextile, smart textile, and medical textiles that ensure active agent delivery across cell membranes [94].

The stimuli-responsive liposome can slowly and controllably release the encapsulated substances. If they have been bonded to textile functional groups with weak bonds, different stimuli may evoke detaching them from textile and crossing of skin layers to deliver the active substance to the desire target. Another possible action is releasing only active substances in vitro and acting as promoters of their penetration through the upper skin layers. The stimuli can be either internal (e.g., enzyme activity, pH changes, or availability of reducing agents) or external (e.g., temperature, light, magnetic field, or ultrasound) [95].

Because liposomes mimic skin composition, they act as effective carriers to enhance the transdermal drug release. Two kinds of lipids (internal wool lipids IWL or phosphatidylcholine PC) have been used for preparation of liposomes and applied onto cotton and polyamide fabrics by exhaustion treatments. PC liposomes exhibit more affinity to both fabrics than IWL liposomes. It has been found that, polyamide fabric adsorbs a slightly higher percentage of liposomes and its release of the encapsulated compound to the skin is higher than that of cotton fabric. The authors have also concluded that biofunctional textiles support skin penetration of drug formulation when only a formulation is being applied to the skin [96]. Liposomes are both chemically and physically unstable [97]. Not always an optimal amount of BAS could be delivered using them. Moreover, an unwanted release of hydrophobic substances can occur, not to mention their high cost. However, many researchers are currently working intensively to overcome these shortcomings [98]. Yamazaki et al. have synthesized new functional liposomes modified with copolymers based on methacrylate. Three kinds of monomers: methoxy diethyleneglycol methacrylate (MD), methacrylic acid (MAA), and lauroxy tetraethyleneglycol methacrylate (LT) were used to prepare copolymers poly (MD-co-MAA-co-LT), that change their solubility in response to both pH and temperature. These liposomes can be used in cosmetic and transdermal delivery systems [99]. The observed problems with liposomes as drug delivery carriers have been overcome by development of various liposome-like vesicles with penetration enhancement properties, such as transfersomes, ethosomes, niosomes, and invasomes [100].

### 7.4. Polymeric Nanoparticles

Polymer nanoparticles are also suitable BAS carriers of therapeutic dose for effective delivery. The requirements for non-toxicity, biocompatibility, and biodegradability are met by: poly (lactide-co-glycolide), poly (lactic acid), poly (ɛ-caprolactone) (PCL), chitosan, gelatin, etc., which have been studied intensively in recent years [101,102,103,104,105].

PCL is a synthetic polymer that is biodegradable in body fluids due to the rupture of its ester bonds. Because of its biocompatibility, it finds various biomedical applications: in tissue engineering, for implantable devices, in nanomedicine, in BAS delivery systems [106,107]. The release profile of the encapsulated BAS in this case depends on the mechanism of polymer degradation and on the properties of the bioactive substance itself, which influences its distribution in the structure of polymer nanoparticles. Lipophilic substances are usually evenly distributed in the polymer matrix, while hydrophilic ones tend to move to the surface of PCL in the adsorbed state. Therefore, such systems release BAS in a two-phase model in which the process rate is much higher for the hydrophilic substances than for the lipophilic ones. During the formulation processes, the hydrophilic BAS accumulates on the surface of the nanoparticles and is released by desorption in the initial period of application. In the case of lipophilic BAS, diffusion is much slower or almost non-existent, and then they are released by surface erosion caused by the enzymatic action.

Nanoparticles from poly-ε-caprolactone with encapsulated hydrophilic BAS caffeine have been produced by a flash nanoprecipitation technique. Nanoparticles should have a suitable size to permeate through the skin (approximately 450 nm). Two kinds of fabrics (knitted cotton fabrics single jersey made of 100% cotton and a blend of viscose/micromodal 70/30% one) have been functionalized by imbibition. The combination of nanoparticles with a proper textile material supports the effective drug release control. The new material possesses both the antioxidant properties of the caffeine and wearability and prolonged action without the need for further patient effort. It is unlike creams and ointments that must be locally applied several times a day [66]. The same method has been used for encapsulation of melatonin and modification of cotton fabric. SEM analysis confirmed uniform distribution of the nanoparticles over cotton fibers. It has been found that functionalized fabrics can be an effective device for delivering melatonin through the skin in a continuous and controlled way [108].

### 7.5. Hydrogels

Hydrogels are three-dimensional polymer networks that absorb large amounts of water or biological fluids. Modern pharmaceutical science pays more and more attention to them as potential BAS carriers. They are suitable for the purpose mainly owing to their physicochemical and biological properties, such as swelling and shrinkage, good sorption capacity, possibility for controlled release of BAS by appropriate change in the conditions, good biocompatibility, good oxygen permeability, low interfacial tension, non-toxicity, etc. In addition, they offer several other characteristics such as bioadhesion, mucoadhesion, easy surface modification corresponding to the shape of the surface over which are being applied [107]. The biocompatibility of hydrogels is determined by their higher water content, hence they are close in morphology and compatible with most natural biological tissues. Another advantage is that they can be made of polymers that decompose into harmless products.

The main disadvantage of hydrogels is their poor mechanical strength, so their interaction with textiles would provide stability and the possibility of long-term use. Photopolymerization has turned out an interesting way to obtain a hydrogel with tailored properties. The physicochemical properties of liquids and solids change under the action of light, laser radiation, etc. Visible or UV light interacts with compounds called photoinitiators, which form free radicals that initiate polymerization to form polymers or hydrogels [109]. Photopolymerization has several advantages over conventional polymerization. Some of them are the possible spatial-temporal control and high rate (from a few seconds to several minutes) of the polymerization, mild synthesis conditions (at room or physiological temperature) with minimal heat release. This makes it suitable for application for materials sensitive to high temperature [110].

Currently pH and thermosensitive hydrogels have emerged and are used for controlled transport and release of BAS in certain organs. They refer to the so-called “smart polymers”. Hydrogels based on copolymers of acrylamide and N-vinylpyrrolidone with alkenic acids and their esters have been used because they are hydrophilic, biocompatible, and non-toxic [111,112,113,114]. Various antifungal and anti-inflammatory agents based on hydrogels have been developed using different polymers [115,116]. For the topical application of triclosan in the treatment of acne, adhesive hydrogel patches based on sodium polyacrylate and carboxymethyl cellulose have been studied [117]. Hydrogels have been found to be successful in the treatment of burn wounds, ensuring healing, which is faster in humid than in a dry environment. The presence of an antimicrobial substance further protects the wound from infection and accelerates the healing process [118,119].

Many skin diseases, injuries, and transdermal therapy require a holistic approach. For example, the hydrogel provides hydration of the skin combined with the supply of a drug, improving the condition of patients with atopic dermatitis. A thermosensitive hydrogel (poloxamer 407/carboxymethylcellulose sodium) with dual functions for moisturizing the skin and drug delivery has been developed [120]. The authors report a hydrogel composition suitable for transdermal delivery of a herbal drug exhibiting antibacterial and anti-inflammatory activities. The compound is also suitable for the production of hydrogel-coated fabrics, as shown in Figure 4. Chitosan modified temperature responsive poly(N-isopropylacrylamide)polyurethane (PNIPA Am/PU) hydrogel has been applied on nonwoven fabric. The hydrogel may be loading with a variety of nutrients (or other functional components), which can be released at body temperature. The obtained material can be reusable with repeated rinsing [44]. Mocanu et al. have prepared bioactive textiles that had been obtained by coating a cotton fabric or chemically modified cotton (having aldehyde or carboxymethyl functional groups) with chitosan either upon crosslinking with glutaraldehyde or as fine particles (after crosslinking with natrium tripolyphosphate). Next, the materials have been treated with BAS that release depending on the type of fabric coating. These bioactive textiles have specific biological activity and could be used for clothes for people with allergies or other skin problems [121].

A thermoresponsive hydrogel has been synthesized through coupling of poly (ethylene glycol) and poly (ε-caprolactone) with hexamethylene diisocyanate as a chemical linker. The prepared hydrogel has been loaded onto a textile material by physical forces. Beforehand, it was coated with a copolymer dissolved in water onto the surface of nonwoven fabric pad, made of rayon and polyester. Skin temperature influences the transition between hydrophilicity and hydrophobicity of the hydrogel. The resulting composite material has been tested for applications in skincare and wound healing. It has demonstrated biocompatibility, good hydration, exudate management, and drug release control of aloin (aloe extract) or curcumin (turmeric extract) [122]. The aqueous dispersion of microgel particles and crosslinking agent citric acid has been applied on cotton fabric and fix by the pad-dry-cure method. Compared with pure cotton, the microgel-loaded fabrics exhibit apparent drug release behavior in response to temperature and the amount of aloin released increases upon heating above the LCST of copolymer [123].

## 8. Multifunctional and Intelligent Textiles—Perspectives and Expectations

Table 1 summarizes some examples for functionalization of textile materials with different drug carriers and their possible applications.

The development of systems for the delivery of therapeutic compounds and in particular for transdermal therapy is associated with the widespread use of biomacromolecules from renewable sources and natural origin (plant or animal). These are proteins (collagen, keratin, fibroin, etc.), lipoproteins, and polysaccharides (chitosan, cyclodextrin, hyaluronic acid, heparin, pectin, etc.). Their advantages compared to synthetic compounds are non-toxicity, non-immunogenicity, good biocompatibility. They have different functional groups (hydroxyl, amino, and carboxyl), which can be modified and thus be permanently linked to the textile materials.

The development of biofunctional textile materials that release BAS in a controlled manner, allowing monitoring the process via a change in their properties is an example of creating new intelligent materials. Those materials can react to various physical, chemical, or biological environment factors (e.g., temperature, pH, light, magnetic field, solvent, etc.). Other impacts are the appearance or increase in the concentration of various biological products (enzymes, elevated glucose levels, etc.) [124,125]. The design of such materials should combine their multifunctional action with monitoring the health status and/or the effects of various environmental stimuli, followed by an appropriate response associated with a change in their properties, to the delivery of BAS, to antimicrobial and other reactions related to supporting the healing process.

## 9. Conclusions

In this review, an analysis of medication used for skin topical and transdermal treatment in medical and clinical practice has been made. In this regard, the structure of human skin, the mechanism for transdermal BAS delivery, and the factors determining the delivery efficiency are described. It has also been shown that BAS can be applied directly to the textile matrix, but its delivery control can be achieved upon their encapsulation or bonding to a suitable carrier. The promising and intensively studied stimuli-responsive drug carriers as dendrimers, micelles, liposomes, nanoparticles, and hydrogels are described in detail. On the basis of literature, it is has been shown that, bioactive composites involving textiles are new perspective biomaterials for the preparation of medical devices as wound dressings, patches, BAS carriers in transdermal therapy, in degenerative diseases, etc. The combined traditional properties of textile with controlled BAS delivery, and the ability to trace visually the drug delivery process will provide information, convenience, and quick response in a given situation. This will certainly lead to improved human health and social status.

## Figures and Tables

**Figure 1 materials-14-00930-f001:**
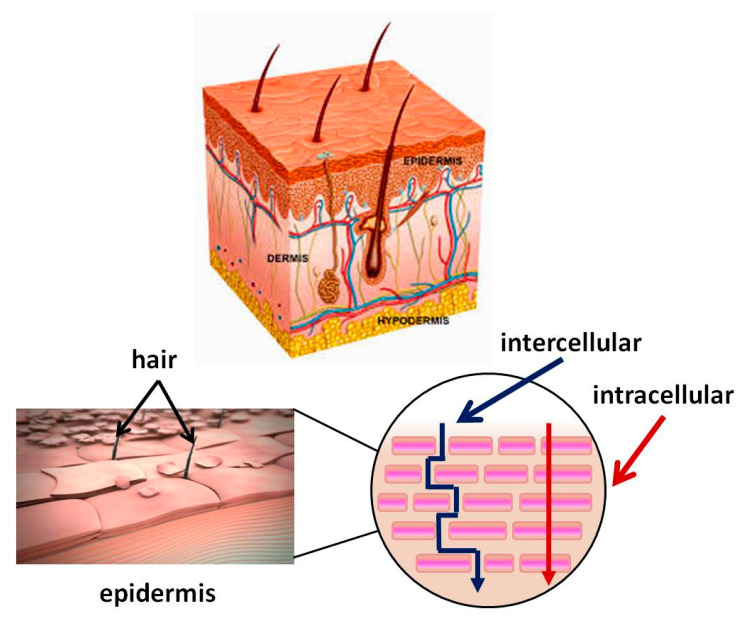
Skin structure. Mechanism of passage through the skin of biologically active substance BAS.

**Figure 2 materials-14-00930-f002:**
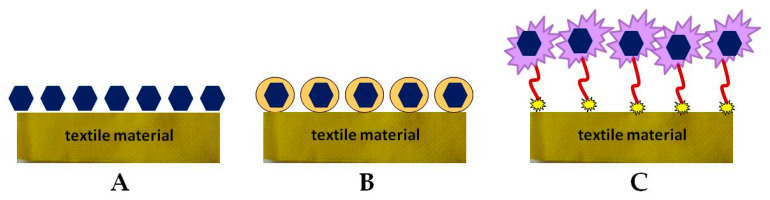
Methods for immobilizing bioactive compounds (BAS) onto a textile material: (**A**) direct immobilization; (**B**) encapsulation; (**C**) via a spaser.

**Figure 3 materials-14-00930-f003:**
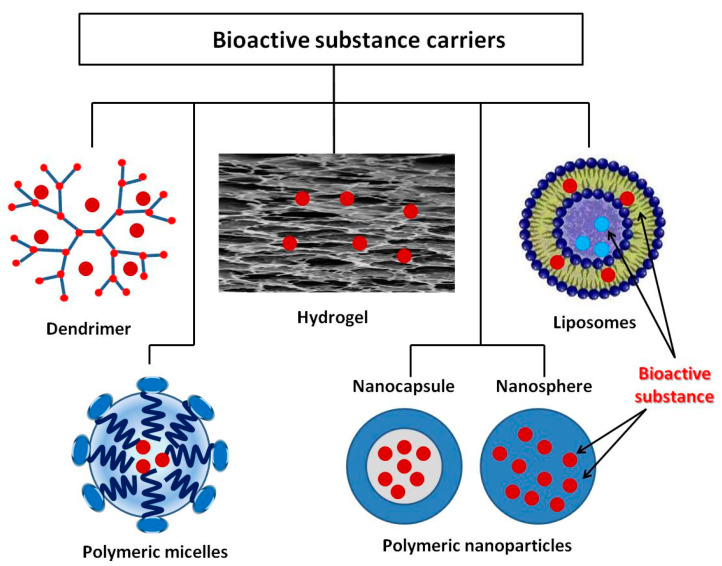
BAS delivery carriers.

**Figure 4 materials-14-00930-f004:**
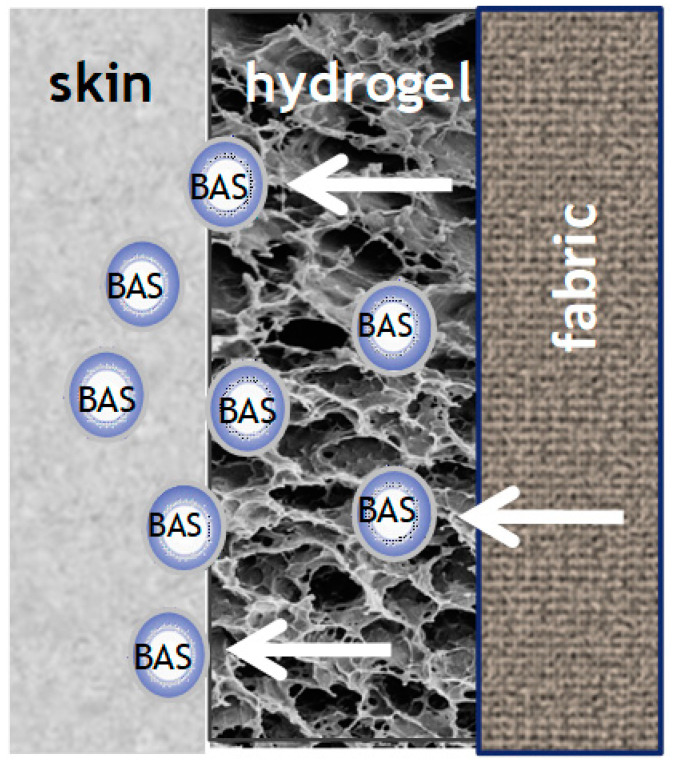
Release of BAS from a hydrogel immobilized onto a modified textile.

**Table 1 materials-14-00930-t001:** Characteristics, functionalization, and application of biofunctional textiles.

Textile Chemical Composition and Structure	Drug Carrier	Textile Functionalization	Application	Ref.
nonwoven cotton fabric	Chitosan modified temperature responsive poly(N-isopropylacrylamide)polyurethane (PNIPA Am/PU) hydrogel	copolymerization on the surface of fabric	loading a variety of nutrients (or other functional components), which can release at body temperature	[44]
knitted cotton fabrics single jersey made of 100% cotton and a blend of viscose/micromodal 70/30%	Nanoparticles from poly-ε-caprolactone with encapsulated hydrophilic BAS caffeine	imbibition	antioxidant properties of the caffeine in transdermal delivery	[66]
cotton and polyamide fabrics	Liposomes made of two kinds of lipids (internal wool lipids or phosphatidylcholine)	exhaustion	transdermal delivery	[95]
cotton fabrics	Nanoparticles from poly-ε-caprolactone with encapsulated hydrophilic BAS melatonin	imbibition	melatonin in transdermal delivery	[108]
cotton fabric or chemically modified cotton (having aldehyde or carboxymethyl functional groups)	chitosan either upon crosslinking with glutaraldehyde or as fine particles (after crosslinking with natrium tripolyphosphate).	coating	clothes for people with allergies or other skin problems, due to the specific biological activity	[121]
nonwoven fabric pad, made of rayon/polyester	thermoresponsive hydrogel poly (ethylene glycol) and poly (ε-caprolactone) with hexamethylene diisocyanate	coating	skincare, wound healing, drug release control of aloin and curcumin	[122]
cotton fabric	thermosensitive microgels copolymer of poly(N-vinylcaprolactam) and chitosan oligosaccharide	pad-dry-cure and attachment by citric acid	textile-based drug delivery system for treating sunburn or skin care	[123]

## Data Availability

Data sharing not applicable.

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
