# Peer review of "Textile Materials Modified with Stimuli-Responsive Drug Carrier for Skin Topical and Transdermal Delivery"

_materials, 2021, doi:10.3390/ma14040930_

Round 1
Reviewer 1 Report
Dear Authors
the text lacks of the explanation of what textiles are from the chemical point of view and which can be used in the transdermal delivery At the beginning of the introduction you should provide a table with each example link to their applications associated to references. The Introduction is not well structured as the Authors listed a large fields of applications without a comprehensive description nor of the textiles neither of the systems in which textiles are present. The Introduction should be deeply reorganized focusing on what is present in literature among the application of textiles in the transdermal delivery. I don't understand the application of a textile in a pharmaceutical dosage form, are there any?
In general the text provides often not really proper references (for example ref 5) or they are even completely absent.
There are also a few typing errors (lines 157,184,185).
line 44 please insert references for each example of smart textiles' utilization listed or provide relevant studies which can support your opinion
line 52 and line 55 please add references about the application of textiles from the 20 th century till now in the medical field
line 56 this sentence is out of the aim of the review and it can be deleted
from line 58 to 65 again the text lacks of many mandatory references
line 62 which are the textiles used in aromatheraphy, antimicrobials, painkillers etc....? You should provide the examples of these systems and devices, describing them in a few sentences
line 66-74 this paragraph should be inserted before the applications
Subchapter 2
The description is really too approximate. which are the textiles used in patches for transdermal delivery? You mentioned the patches, but you know that there are four generations of developped patches? You should mention them at least. Here, the references are again missing.
Subchapter 3
The transdermal delivery isn't only affected by skin composition but also by drug characteristics (PM, log P, pharmacological activity or its active dose), you don't mention these aspects, please provide a more complete description of this topic
Subchapter 4
line 140-148 please provide a table in which insert all the systems you mentioned present in the literature and add a brief description of each in the text
line 144 please add references
Subchapter 5
The Authors should describe only the requirements of biofunctional textiles and not those related to delivery systems. This confused approach is present along all the text but the title focuses on textiles and not about delivery systems. The result is that the topics are always approximatively described as there should be many more things to say.
Subchapter 6
The Authors should report and describe more in depth examples of denderimers, liposomes, polymeric micelles etc... used in biofunctional textiles otherwise other applications as such for drug delivery in medical field are out of topic.
Therefore I suggest the Authors to reorganize the text focusing more on the topic, to report the examples found in the literature in at least one table before resubmitting the review. A review without a table and with only 45 references implies that the work done was too fast.
Author Response
We would like to thank the reviewer for his constructive remarks, which led to an improvement of the manuscript compared to the original version. In this regard, much of the text was rewritten and new text and literature sources were introduced.In general the text provides often not really proper references (for example ref 5) or they are even completely absent.
Ref.5 has been changed
There are also a few typing errors (lines 157,184,185).
corrected
line 44 please insert references for each example of smart textiles' utilization listed or provide relevant studies which can support your opinion
references have been added
line 52 and line 55 please add references about the application of textiles from the 20 th century till now in the medical field
added
line 56 this sentence is out of the aim of the review and it can be deleted
from line 58 to 65 again the text lacks of many mandatory references
The references have been added
line 62 which are the textiles used in aromatheraphy, antimicrobials, painkillers etc....? You should provide the examples of these systems and devices, describing them in a few sentences
the respective examples have been added
line 66-74 this paragraph should be inserted before the applications
the paragraph has been replaced
Subchapter 2
The description is really too approximate. which are the textiles used in patches for transdermal delivery? You mentioned the patches, but you know that there are four generations of developped patches? You should mention them at least. Here, the references are again missing.
The explanation was added in the text
Subchapter 3
The transdermal delivery isn't only affected by skin composition but also by drug characteristics (PM, log P, pharmacological activity or its active dose), you don't mention these aspects, please provide a more complete description of this topic
This description has been done I the text.
Subchapter 4
line 140-148 please provide a table in which insert all the systems you mentioned present in the literature and add a brief description of each in the text
The table was inserted.
line 144 please add references
Done
Subchapter 5
The Authors should describe only the requirements of biofunctional textiles and not those related to delivery systems. This confused approach is present along all the text but the title focuses on textiles and not about delivery systems. The result is that the topics are always approximatively described as there should be many more things to say.
In this case first, we have given information on the suitability of such materials as delivery systems and then to show their use in transdermal therapy after their deposition on textile fabrics.
Subchapter 6
The Authors should report and describe more in depth examples of denderimers, liposomes, polymeric micelles etc... used in biofunctional textiles otherwise other applications as such for drug delivery in medical field are out of topic.
The text was rewritten according to the requirements of the reviewer
Reviewer 2 Report
If the authors of this paper were not editors of this special issues, I would have decided to “reject”. Since the topic of this manuscript ‘Textile materials in transdermal therapy’ is very interesting, I read it very carefully from start to end. However, this manuscript has no academic information and no intensive discussion. The most serious of all is that unlike the title, there is very little about the ‘textile materials’ for transdermal therapy, only listing of basic polymer DDS systems, such as dendrimers, micelles, NPs etc. Since the quality of the just a few figures is also very low (figure 2, 3, 4), messages can not be conveyed. So, to become a review paper that fits the special issue, authors must rewrite the paper completely.
Author Response
We would like to thank the reviewer for his constructive remarks, which led to an improvement of the manuscript compared to the original version. Much of the text has been rewritten and some new paragraphs have been introduced. Special attention was paid to the influence of textile materials in transdermal and skin topical therapy, and relevant literature sources were introduced.
Reviewer 3 Report
However, for this moment the review requires significant improvement.
More references are needed. For example, section “5. Requirements to be met by BAS delivering systems and bifunctional textiles” is an important part of the work but has only one reference. In the end of the section 4 there is a statement “Bifunctional textile materials can be produced by traditional techniques: exhaustion from solution; irrigation; spraying; pad-dry-cure; using sol-gel technique”, which also requires references. Moreover, the technologies allowing to fabricate bifunctional materials also include plasma treatment methods, chemical modification strategies and other approaches, which should be mensioned. Also, it is a rapidly developing field of science, there are a lot of new relevant articles, so authors need to update the references. For example:
- Line 177. “In recent years, however, active studies have been carried out on other structures…” and then authors refer to 2002 article (ten Breteler, M.; Nierstrasz, V.; Warmoeskerken, M. Textile slow-release systems with medical applications. AUTEX Research Journal. 2002, 2, 175-189.).
Figure 2. Methods for immobilizing bioactive compounds onto a textile material. This is an incomprehensible figure. Explanations should be presented either in the text or in the figure.
The authors claim that aromatherapy is an area where textiles are involved as an important functional element of systems for transdermal therapy twice in the article (line 15, line 62) without any references. Please explain this.
In addition, in my opinion, it is needed to add a paragraph on the comparison of the textiles for transdermal therapy with other types of biomaterials utilized in that sphere (e.g. microneedles, ointments, etc.).
English language should be improved, numerous spelling and grammar mistakes were found, there are some of them:
- Line 19. “… method is by microencapsulation”, “by” is the an extra word in a sentence;
- Line 19(20). “…stimuli responsive”, “stimuli-responsive” is spelled with a hyphen, not separately;
- Line 25. “…structures which can be used”, it is more correct to use “that” rather than “which”;
- Line 36. “…population what largely depends on”, it is more correct to use “that” rather than “what”;
- Line 36. “…implemening”, word “implementing” spelled like this;
- Line 60. “…deficiencies, what leads to”, it is more correct to use “that” rather than “what”;
- Line 72. “Utilization of …”, the utilization;
- Line 81. “… aspect in medical…”, aspect off;
- Line 94. “… and tratment costs…”, word “treatment” spelled like this;
- Line 113. “… in many deposites…”, correct writing is “deposits”;
- Line 114(115) “… corneum is water resistant…”, ” water-resistant” is spelled with a hyphen, not separately;
- Line 118. “Hydrophobic drugs penetrate the skin intercellularly.” The word “intercellularly” is wrong, it is more correct to use “intercellular”, “the” is an extra word;
- Line 178. “…delivery of drugs, what is…”, it is more correct to use “that” rather than “what”;
- Line 181. “… treating of acne…”, “of” is an extra word in a sentence;
- Line 184. “… detail bellow”, “bellow” and “below” are words with different meanings, in this case it is necessary to use “below”;
- Line 196. “… water solubility”, “water-solubility” is spelled with a hyphen, not separately;
- Line 238. “… itself, what influences”, it is more correct to use “which” rather than “what”;
- Line 262. “Photopolymerization has turned to be…”, “turned out”;
- Line 263. “The the physicochemical…”, one “the”;
- Line 277. “For topical…”, “For the topical”;
- Line 298. “Other impacts are appearance…”, “Other impacts are the appearance…”;
- Line 303. “… antimicrobiality and…”, “antimicrobial”;
- Line 311. “… in the conditions can lead…”, “… in the conditions that can lead…”.

Author Response
More references are needed. For example, section “5. Requirements to be met by BAS delivering systems and bifunctional textiles” is an important part of the work but has only one reference. In the end of the section 4 there is a statement “Bifunctional textile materials can be produced by traditional techniques: exhaustion from solution; irrigation; spraying; pad-dry-cure; using sol-gel technique”, which also requires references. Moreover, the technologies allowing to fabricate bifunctional materials also include plasma treatment methods, chemical modification strategies and other approaches, which should be mensioned. Also, it is a rapidly developing field of science, there are a lot of new relevant articles, so authors need to update the references. For example:
Much of the manuscript has been revised according to your comments and additional literature sources have been introduced.
- Line 177. “In recent years, however, active studies have been carried out on other structures…” and then authors refer to 2002 article (ten Breteler, M.; Nierstrasz, V.; Warmoeskerken, M. Textile slow-release systems with medical applications. AUTEX Research Journal. 2002, 2, 175-189.).
- Some new references have been used.
Figure 2. Methods for immobilizing bioactive compounds onto a textile material. This is an incomprehensible figure. Explanations should be presented either in the text or in the figure.
The explanations related to Figure 2 are given in the text and in the caption to the figure.
The authors claim that aromatherapy is an area where textiles are involved as an important functional element of systems for transdermal therapy twice in the article (line 15, line 62) without any references. Please explain this.
Some appropriately references have been citrated
In addition, in my opinion, it is needed to add a paragraph on the comparison of the textiles for transdermal therapy with other types of biomaterials utilized in that sphere (e.g. microneedles, ointments, etc.).
Some comments and references for comparison of transdermal therapy and other types of biomaterials utilized in that sphere have been added.
English language should be improved, numerous spelling and grammar mistakes were found, there are some of them:
All spelling and grammar mistakes have been corrected. All spelling and grammar mistakes have been unproved
Reviewer 4 Report
Comments and remarks:
- the title, please change it, please add the instrumental/analytical techniques used,
- in the introduction part should be highlighted the main aim of the paper, and additionally, what is the novelty of carried research work,
- what about thermoresponsive hydrogels and their biomedical applications?
- what about pharmaceutical (nano)carries?
Conclusions: Minor revision.
Author Response
We thank the reviewer for the recommendations which were taken into account in the revision of the text.
the title, please change it, please add the instrumental/analytical techniques used,
The title has been changed and ant the instrumental and analytical methods and techniques were added to the text.
- in the introduction part should be highlighted the main aim of the paper, and additionally, what is the novelty of carried research work,
The introduction was improved
- what about thermoresponsive hydrogels and their biomedical applications?
The use of thermoresponsive hydrogels and their biomedical applications have been added to the text.
- what about pharmaceutical (nano)carries?
The use of nanocarriers in transdermal therapy was explained in the text.
Round 2
Reviewer 1 Report
The Authors have satisfied all my requestsby significantly improving their contribution which in this form is ready to be published.
Author Response
We thank the reviewer for the overall opinion and comments during the evaluation of the manuscript.
Reviewer 2 Report
The authors address the comments. So I agree to the publication of this manuscript.
Author Response

(The authors gave the same response as above.)

Reviewer 3 Report
The authors have done a tremendous amount of work to improve the quality of the reviews. The authors were attentive to the comments. Many relevant links have been added, additional sections have been written.
Some comments and references for comparison of transdermal therapy and other types of biomaterials in that sphere have been added. I asked to add a comparison with ointments, this was not done. This remark may remain for the editor's decision.
Also, due to the addition of new material, a very large number of grammatical errors and misspellings were added, which greatly complicates the reading and understanding of the work. Errors must be eliminated.
If the authors take into account remarks and comments, then the proposed article will become a useful tool for a scientific group around the world.
Author Response
The authors have done a tremendous amount of work to improve the quality of the reviews. The authors were attentive to the comments. Many relevant links have been added, additional sections have been written.
Some comments and references for comparison of transdermal therapy and other types of biomaterials in that sphere have been added. I asked to add a comparison with ointments, this was not done. This remark may remain for the editor's decision.
The comparison with ointments, has been done in the text.
Also, due to the addition of new material, a very large number of grammatical errors and misspellings were added, which greatly complicates the reading and understanding of the work. Errors must be eliminated.
Grammatical and spelling errors were corrected in the text.
If the authors take into account remarks and comments, then the proposed article will become a useful tool for a scientific group around the world.